# Molecular Characteristics and Antibiotic Resistance of *Staphylococcus aureus* Isolated from Patient and Food Samples in Shijiazhuang, China

**DOI:** 10.3390/pathogens11111333

**Published:** 2022-11-11

**Authors:** Han Zhang, Liyun Qin, Caiping Jin, Huidong Ju, Ruiping Jiang, Lijie Li, Hong Zhang, Weili Gao, Xiuping Wei, Hongyan Dong, Fei Lu, Guoping Lv

**Affiliations:** 1Basic Medicine College, Hebei University of Chinese Medicine, Shijiazhuang 050200, China; 2Shijiazhuang Center for Disease Control and Prevention, Shijiazhuang 050011, China; 3Hebei Key Laboratory of Unidentifiable Bacteria, Shijiazhuang 050011, China; 4College of Chemical Technology, Shijiazhuang University, Shijiazhuang 050035, China

**Keywords:** *Staphylococcus aureus*, MLST, clonal complex, antibiotics resistance, MRSA, MSSA

## Abstract

*Staphylococcus aureus (S. aureus)* is a common opportunistic and zoonotic pathogen in the world and could easily cause human infections and food contaminations. This study investigated the sequence typing and resistance profiles of *S. aureus* isolates from patient and food samples in Shijiazhuang, China. A total of 101 *S. aureus* isolates were distributed into six clonal complexes (CCs) and 16 singletons. A total of 86 patient isolates were distributed into six clonal CCs and 12 singletons, including a new ST. CC59, CC5, CC22, and CC398 were the predominant CCs of patient isolates. A total of 15 foodborne *S. aureus* isolates were distributed into 3 CCs and 4 STs, and CC1 was the most prevalent CC. Moreover, 101 *S. aureus* isolates had high resistance to penicillin and low resistance to chloramphenicol and rifampicin. A total of 39 strains of methicillin-resistant *Staphylococcus aureus* (MRSA) were detected in this study, including thirty-eight strains of patient isolates (44.2%, 38/86) and one strain of food isolates (6.7%, 1/15). MRSA-ST5, MRSA-ST59, and MRSA-ST239 were the predominant MRSA isolates in hospitals. The present study explained the relationship between *S. aureus* isolated from patient and food samples and indicated the risks of *S. aureus* in infectious diseases.

## 1. Introduction

As a common zoonotic pathogen, *S. aureus* can cause serious diseases in humans [1,2]. *S. aureus* is one of the main pathogenic bacteria of infectious diseases, and can cause food poisoning outbreaks [1,3]. *S. aureus* from patient samples mainly comes from infectious diseases and is responsible for a wide variety of community-acquired or hospital-acquired infections [4,5,6]. It often contaminates foods and produces a wide variety of toxins, and contamination increases the risk of infectious diseases and food poisoning outbreaks [2,7]. *S. aureus* can cause poultry myelitis, sepsis, mastitis in cows and other animals, and contamination dairy products [8,9,10]. *S. aureus* transmission could contaminate environments and food, and endanger human and animal health [8,11]. *S. aureus* isolated from patient and food samples are closely related to human health.

*S. aureus* is prone to antimicrobial resistance and poses a great challenge in the treatment of related infectious diseases [12,13,14]. It presents high resistance to penicillin [15,16], and many *S. aureus* isolates can be resistant to multiple antibiotics. Methicillin-resistant *S. aureus* (MRSA) strains are resistant to most semisynthetic penicillin, and have various resistance to other antibiotics [6,14,16,17]. Compared with methicillin-susceptible *S. aureus* (MSSA), MRSA causes longer length of hospital stay, higher infection rates and higher mortality [6,18]. MRSA is distributed all over the world and poses a great threat to global public health.

The epidemic characteristics of *S. aureus* are important for understanding the transmissions and pathogenicity in different parts of the world. Multilocus sequence typing (MLST) is a general analytical tool that is used for molecular traceability analysis of bacteria worldwide. It is used at a global scale to compare and analyze a pathogen in different countries and different periods. The molecular characteristics of *S. aureus* vary with different regions and sources. In Shijiazhuang, China, the epidemic sequence types (STs) of *S. aureus* in foods are ST1, ST5, ST59, ST398, ST7, and ST9 [3]. Those STs are also prevalent in Chinese retail meat products [19]. ST59, ST9, ST1, and ST398 are the main STs of food-related MRSA in China [20]. However, the prevalent STs of clinical *S. aureus* are different from foodborne *S. aureus* in China.

The predominant STs of clinical *S. aureus* are ST5, ST188, ST239, ST59, ST7, and ST398 in teaching hospitals [21]. ST5, ST239, and ST59 are the prevalent clinical MRSA isolates in China [17,21]. ST9, ST398, and ST5 were predominant genotypes in USA [22], and ST93 was the most common MRSA of clinical isolates in Australia [23]. *S. aureus* isolates have different epidemic molecular types at different times, even if they come from the same source [21]. Among clinical *S. aureus* in Hainan, ST45, ST188, ST59, and ST88 were the prevalent STs from 2013 to 2014, but ST398, ST188, and ST59 were prevalent from 2019 to 2020 [16]. Therefore, MLST can help to monitor the molecular characteristics of *S. aureus* in food contaminations and bacterial infections.

The aim of this study was to characterize *S. aureus* isolated from patients and food samples in Shijiazhuang, China. A total of 101 *S. aureus* isolates were collected from hospitals and foods in 2019. In particular, MLST genotypes and resistance profiles were determined. In addition, the MRSA isolates were detected.

## 2. Methods

### 2.1. Bacterial Isolates

This study examined 101 *S. aureus* isolates from different tertiary hospitals and foods in 2019 from Shijiazhuang, China. There were 15 foodborne isolates from 192 food samples under surveillance for microbiological risk, including raw meat, raw poultry, aquatic products, cooked food, milk powder, and nutritious rice flour. Among them, there were seven isolates of *S. aureus* detected in raw meat, three isolates in raw poultry and five isolates in cooked meat. The 86 patient isolates included sputum sputa (*n* = 34, 39.5%), pus and secretion (*n* = 13, 15.1%), blood (*n* = 4, 4.7%), swabs (*n* = 11, 12.8%), and others (*n* = 24, 27.9%). Routine identification methods for *S. aureus* were conducted, including Gram staining, catalase tests, coagulase tests, and Matrix-Assisted Laser Desorption Ionization Time of Flight Mass Spectrometry (MALDI-TOF MS) System (Bruker, Berlin, Germany). The *S. aureus* strain ATCC25923 was used as a control. All isolates were then frozen and maintained in a Brain Heart Infusion (BHI) medium with 40% glycerol at −80 °C for further experiments.

### 2.2. Extraction DNA and Detection of Antibiotic Resistance Genes

Bacterial DNA was extracted by a bacterial genomic DNA extraction kit (DNeasy Blood and Tissue Kit, Qiagen Inc, Redwood City, CA, United States), and the *blaZ*, *mec* and *nucA* genes were detected using PCR assays. The *nucA* gene was used as an internal reference gene. All of the primers (TaKaRa, Beijing, China) were listed in our previous study [3]. A multiplex PCR assay was used to evaluate the reference gene *nucA* and to detect whether isolates carried antimicrobial resistance genes *blaZ* and *mec*. The PCR mixture contained 0.6 µL *mec*-F (10 µM), 0.6 µL *mec*-R (10 µM), 0.5 µL *blaZ*-F (10 µM), 0.5 µL *blaZ*-R (10 µM), 0.4 µL *nucA*-F (10 µM), 0.4 µL *nucA*-R (10 µM), 4 µL genome DNA template, 12.5 µL of premixTap TaKaRa (TaKaRA, China), and double-distilled water added to obtain a volume 25 µL. The PCR conditions were as follows: initial denaturation at 95 °C for 4 min, followed by 35 cycles of denaturation at 95 °C for 30 s, annealing at 54 °C for 30 s, extension at 72 °C for 1 min 40 s, and a final extension at 72 °C for 7 min. The PCR isolation method and the interpretation of the results come from our previous studies [3].

### 2.3. Antimicrobial Susceptibility Test

The Kirby-Bauer paper dispersion method was used for an antimicrobial susceptibility test. A total of 14 antibiotics were tested, including cefoxitin (FOX, 30 μg), penicillin G (PEN, 10U), oxacillin (OXA, 1 μg), tetracycline (TET, 30 μg), doxycycline (DOX, 30 μg), chloramphenicol (CAP, 30 μg), rifampicin (RFP, 5 μg), erythromycin (ERY, 15 μg), clarithromycin (CLR, 15 μg), sulfamethoxazole (SMZ, 1.25/23.75 μg), gentamicin (GM, 10 μg), clindamycin (CLI, 2 μg), ciprofloxacin (CIP, 5 μg), levofloxacin (LEV, 5 μg). *S. aureus* ATCC 25923 was used as a quality control strain, and results were interpreted according to the guidelines of the Clinical and Laboratory Standards Institute (CLSI M100-S30) [24]. The *mec*-positive or cefoxitin-resistant isolates (cefoxitin minimum zone diameter ≤ 21 mm) were identified as MRSA.

### 2.4. MLST

The base sequence of housekeeping genes of *S. aureus*, (*arcC*, *aroE*, *glpF*, *gmkF*, *pta*, *tpi*, and *yqiL*) were adapted for MLST, which was carried out according to a previous protocol [25]. PCR assays were used to amplify these seven housekeeping genes. The housekeeper genes were sequenced by Sanger dideoxy DNA sequencing (Sangon Biotech, Shanghai, China). The resulting alleles were input into the MLST website (http://saureus.mlst.net accessed on 26 July 2022) and compared with a known sequences search and by locus combinations to determine the allele number, STs, and CCs. The strains that did not match with known STs were named as new STs after submitting them to the database. The clustering of related STs was defined as CCs, and the STs that did not correlate with any clone groups were defined as singletons, which were defined by eBurst.

### 2.5. Statistical Analysis

Statistical analysis was performed using IBM SPSS Statistics 26.0. Pearson’s Chi-square tests or Fisher’s exact tests were used to analyze the data. A phylogenetic tree was constructed using BioNumerics 7.6.2 (Applied Maths, Sint-Martens-Latem, Belgium).

## 3. Results

### 3.1. Antibiotic Resistance Profiles of S. aureus

The 101 *S. aureus* isolates had lower resistance rate to CAP (4.0%, 4/101) and RFP (5.0%, 5/101). The other resistance rates were 97.0% (98/101) for PEN, 68.3% (69/101) for ERY, 68.3% (69/101) for CLR, 67.3% (68/101) for SMZ, 56.4% (57/101) for CLI, 39.6% (40/101) for TET, 38.6% (39/101) for FOX, 37.6% (38/101) for OXA, 26.7% (27/101) for CIP, 21.8% (22/101) for LEV, 17.8% (18/101) for GM, and 14.9% (15/101) for DOX. The difference in resistance rates between patients and food isolates to the same antibiotic is shown in Figure 1.The resistance rates of patient isolates to CAP, RFP, and LEV were 3.5%, 5.8%, and 25.6%, respectively, but the resistance rate of food isolates to CAP was 6.7%, and they were completely sensitive to RFP and LEV.

The patient isolates had significantly higher resistance rates to FOX than the food isolates (44.2% vs. 6.7%, *p* = 0.008), OXA (43.0% vs. 6.7%, *p* = 0.008), ERY (75.6% vs. 26.7%, *p* = 0.001), CLR (75.6% vs. 26.7%, *p* = 0.000), SMZ (74.4% vs. 26.7%, *p* = 0.001), and CLI (61.6% vs. 26.7%, *p* = 0.022). However, there was no significant statistical difference between patient and food isolates to other antibiotics. The resistance rates of patient isolates were 39.5% for TET, 29.1% for CIP, 25.6% for LEV, 19.8% for GM, 5.8% for RFP, and 3.5% for CAP. In food isolates, the resistance rates to these antibiotics were 40.0%, 13.3%, 0%, 6.7%, 0%, and 6.7%, respectively.

The 101 *S. aureus* isolates were examined for the *mec* and *blaZ* gene. A total of 38.6% (39/101) of isolates were positive for *mec* gene, and 76.2% (77/101) were positive for the *blaZ* gene. The detection rate of the *mec* gene was 44.2% (38/86) in patient isolates and 6.7% (1/15) in food isolates. The detection rates of the *blaZ* gene were 74.4% (64/86) and 86.7% (13/15), respectively.

The 62 MSSA isolates had low resistance rates to DOX (1.6%, 1/62), RFP (1.6%, 1/62), LEV (3.2%, 2/62), and CAP (4.8%, 3/62). However, the resistance rates to PEN, SMZ, ERY, and CLR were higher, at 95.2%, 66.1%, 64.5%, and 64.5%, respectively. The MSSA isolates included 48 patient isolates and 14 food isolates. The patient MSSA isolates were resistant to PEN (95.8%, 46/48), SMZ (79.2%, 38/48), ERY (77.1%, 37/48), and CLR (77.1%, 37/48), but no *S. aureus* isolate was resistant to DOX. The food MSSA isolates had the highest resistance rate to PEN (92.9%, 13/14) and were completely sensitive to RFP and LEV. 80.4% (37/46) of the PEN-resistant patient MSSA isolates were *blaZ*-positive strains, while 84.6% (11/13) of the PEN-resistant foodborne MSSA isolates were *blaZ*-positive.

Among the 101 *S. aureus* isolates, 39 isolates were MRSA, and 62 isolates were MSSA. Among the 39 MRSA isolates, thirty-eight were isolated from patient isolates and one was from food isolates. Patients MRSA isolates had higher frequencies than the food MRSA isolates (44.2% vs. 6.7%, *p* < 0.05). The patient MRSA isolates had significantly higher resistance rates to TET (68.4% vs. 16.7%, *p* = 0.000), DOX (36.8% vs. 0.0%, *p* = 0.000), CIP (55.3% vs. 8.3%, *p* = 0.000), LEV (52.6% vs. 4.2%, *p* = 0.000) and CLI (73.7% vs. 52.1%) than the MSSA isolates. (Table 1).

### 3.2. STs and CCs of S. aureus Isolates

A total of 26 STs were detected in the 101 *S. aureus* isolates and were divided into six CCs and 14 STs were singletons. The CCs and singletons included CC1, CC5, CC15, CC22, CC8, CC121, ST59, ST398, ST7, ST3069, ST4845, ST88, ST1281, ST3068, ST20, ST25, ST3355, ST522, ST2472, and a new type. The allelic profile of newly discovered sequence is 7-6-1-5-8-8-1. As shown in Table 2, there were different distribution characteristics between patient isolates and food isolates.

A total of 22 STs were obtained from 86 patient isolates. ST59 was the predominant ST (25.5%, *n* = 22), followed by ST5 (24.4%, *n* = 21), ST22 (11.6%, *n* = 10), ST398 (8.1%, *n* = 7), ST239 (5.8%, *n* = 5), ST15 (3.4%, *n* = 3), ST20 (2.3%, *n* = 2), and ST7 (2.3%, *n* = 2). The other 14 STs (ST121, ST1281, ST2315, ST25, ST306, ST3068, ST3069, ST3355, ST4845, ST6, ST6769, ST72, ST88, and a new ST), each accounted for 1.1% (*n* = 1). These STs belonged to six CCs and 12 singletons. The minimum spanning tree was built from the MLST allelic profiles of 86 *S. aureus* isolates with CCs and singletons (Figure 2). The cladogram was highly diverse. The most prevalent clinical *S. aureus* isolates were CC59, CC5, CC22, and CC398. The other CCs or singletons were isolated sporadically (Figure 2).

A total of 15 food isolates of *S. aureus* from eight STs were assigned to three CCs and four singletons STs, (ST7, ST522, ST59, ST2472, ST1, ST6, ST15, and ST9). CC1, CC5, CC7, and CC59 were the dominant CCs. The predominant CCs or singletons were CC1 (40%, *n* = 6), followed by ST59 (13.3%, *n* = 2), CC5 (13.3%, *n* = 2), and ST7 (13.3%, *n* = 2). The other CCs or singletons were CC15, ST522, and ST2472, which each accounted for 6.7% (*n* = 1).

### 3.3. STs in MRSA and MSSA Isolates

Among the 39 MRSA isolates, thirty-eight strains were from patient isolates, and one strain (MRSA-ST59) was from foodborne isolates. The predominant STs of MRSA isolates in patients were ST5 (15/38, 39.5%), ST59 (12/38, 31.6%), ST239 (4/38, 10.5%), and ST398 (3/38, 7.9%). The other MRSA isolates included ST15, ST25, ST72, and ST3355, which each accounted for 2.6% (1/38). ST59 (19.4%, 12/62) and ST22 (17.7%, 11/62) were the most common in MSSA isolates, followed by ST5 (9.7%, 6/62), ST1 (6.5%, 4/62), ST398 (4.8%, 3/62), ST239 (3.2%, 2/62), ST15 (3.2%, 2/62), ST20 (3.2%, 2/62), ST6 (3.2%, 2/62), ST7 (3.2%, 2/62), and ST9 (3.2%, 2/62). ST121, ST1281, ST2135, ST306, ST3068, ST3355, ST15, ST522, and ST2472 each accounted for 1.6% (1/62).

## 4. Discussion

*S. aureus* isolates were collected from hospitals and food samples for investigation of their antibiotic resistance profiles and molecular characteristics. The isolates had low resistance rate to CAP (4.0%) and RFP (5.0%), while they had high resistance rates to PEN (93.3% for food isolates and 97.7% for patient isolates). The PEN resistance results are consistent with those reported in Hainan hospitals and in food animals in Sichuan, China [16,26]. *S. aureus* isolates are widely resistant to penicillin in China. The antibiotic resistance rate was reported as 84.6% in meat samples, and 85.1% in blood isolates in Shandong [19,27].

The patient isolates were more resistant to ERY, CLR, SMZ, and CLI than the food isolates. The patient isolates had high resistance to ERY (75.6%), CLR (75.6%), SMZ (74.4%), and CLI (61.6%). The resistance was 81.2% for ERY and 78.2% for CLI in Shandong [27]. 95.62% of MRSA isolates and 73.66% of MSSA isolates were resistant to ERY in Shenzhen [28], whereas in Hainan, the resistance rate was 75.0% for ERY and 64.5% for CLI for MRSA isolates, and 35.8% for ERY and 29.8% for CLI in MSSA isolates [16].

In this study, 95.2% (59/62) MSSA isolates were resistant to PEN and 81.4% (48/59) of them carried the *blaZ* gene responsible for resistance to penicillin. MRSA is widely distributed in patient isolates and is rare in food isolates. The food MRSA isolates accounted for 6.7%, which was consistent with isolates from food samples from 2011-2019 in our previous studies [3]. This result was similar to other results from research in China. A total of 6.83% of *S. aureus* isolates from retail foods were MRSA, 7.14% of those from retail meat and meat products in China were MRSA [19,20], and 10.17% of isolates from food animals were MRSA in Sichuan [26].

MRSA is rarely detected in food isolates but is widely distributed in patient isolates. In this study, the proportion of MRSA in patient isolates was 44.2%. It was also 44.2% in Shanghai, but it was 33.5% in Hainan [13,16]. 23.8% of *S. aureus* from blood was MRSA in Shandong [27]. Moreover, MRSA isolates have a more severe resistance than MSSA isolates in terms of a higher resistance rate and a greater number of antibiotic-resistant species [26,28,29].

MLST was performed to analyze the molecular characteristics of the *S. aureus* isolates. The patient isolates were assigned to six CCs and 12 singletons, as shown in Figure 2. The predominant CCs were CC59, CC5, CC22, and CC398. These predominant *S. aureus* CCs were also the dominant blood CCs in Shandong [27]. The dominant CCs have some differences in different regions in China. The dominate CCs in hospitals were CC398, CC59, CC188, and CC45 in Hainan [16], which CC5 and CC398 were the dominant *S. aureus* CCs from bloodstream infections between 2013 and 2018 in Shanghai [13].

The prevalent CCs of patient isolates are different in different countries. CC5, CC8, CC188, CC59, CC7, and CC398 were the most prevalent CCs of *S. aureus* from 22 teaching hospitals in China [21]. In Denmark, the most frequent CCs of *S. aureus* isolates from atopic dermatitis were CC1, CC8 and CC15 [30]. In Poland, CC45, CC30, CC5, and CC15 were the predominant CCs isolated from the oral cavities of dental patients [31]. The majority of the CCs in Kenya were CC152 and CC8 [32]. The prevalence status of *S. aureus* in infectious diseases has important implications for understanding the transmission of the pathogen.

The foodborne *S. aureus* isolates were assigned to three CCs and four singletons STs. The dominant CCs or singletons were CC1, CC5, CC7 and CC59. The most prevalent CCs have not changed in this region. They were also the prevalent CCs in foodborne *S. aureus* isolates during 2011–2019 in our previous study [3]. CC1, CC5, CC7, and CC59 were the common CCs of foodborne *S. aureus* isolates in China [19]. CC59 and CC5 were the predominant clones and are often detected in foods, disease infections, and healthy people in China [19,21,27]. The prevalent CCs in foodborne *S. aureus* isolates vary greatly throughout the world. In Northern Algeria, CC1 and CC97 were prevalent CCs of *S. aureus* from food samples [33]. CC5, CC30, CC15, and CC1 were common in foodborne *S. aureus* in Iowa, USA [29]. In Switzerland, CC705, CC97, CC20, CC479, CC1, CC8, and CC15 were common CCs of *S. aureus* from milk samples [10]. The MLST data can describe the spread of *S. aureus* in foods.

In this study, 22 STs were detected in patient isolates. Among them, ST59, ST5, ST22, and ST398 were the predominant STs (Table 2). ST59, ST5, ST398 were also the common foodborne isolates of *S. aureus* in this study. These are common foodborne STs in China. ST59 is common in retail food samples [20]. ST398 is the main livestock-associated *S. aureus* and the one in retail meat and meat products in China [9,19]. Including both MRSA and MSSA isolates, ST398 is also reported as the predominant clinical isolate in China [16,21,28].

ST5 is commonly isolated from retail meat and meat products, and causes infectious diseases in China [19,21,28]. As a common *S. aureus* isolate in infectious diseases, ST22 was the predominant blood MSSA in Shandong [27], and the main isolate from community-associated skin infections in Beijing, China [34]. ST22 has been one of the most important MRSA clones in infectious diseases in hospitals globally [35,36]. However, it is rarely detected in foods from China [3,19,20].

ST1 was the most predominant ST among the foodborne *S. aureus* isolates in this study. It is also a common food contamination isolate in China [3,19]. ST1 has been isolated from infectious diseases in other regions [16,21,29]. However, it has not been isolated from patients in this study, and no food poisoning outbreaks have been caused by ST1 for the past 10 years in this region [3].

A total of 39 MRSA strains were isolated in this study, including thirty-eight strains from patient isolates and one MRSA-ST59 from foodborne isolates. The predominant STs of MRSA isolates from patient were ST5, ST59, ST239 and ST398. The predominant STs of MRSA isolates are consistent with the epidemiological distributions in China. ST5, ST59, and ST239 were the predominant clinical MRSA isolates in Chinese hospitals [17,18,32] and have been the major epidemic MRSA STs in bloodstream infections by *S. aureus* in China [13,21,27]. MRSA-ST5 was the most predominant MRSA in clinical *S. aureus* isolates in Shijiazhuang, Wuhan, and Shanghai [13,18]. However, MRSA-ST5 has a low distribution in foods [3,20].

As a common MRSA isolate, ST239 is widely distributed in hospitals and is rarely detected in foodborne *S. aureus* [17,18,20,37]. ST239 has been the most predominant MRSA ST in Chinese hospitals [17]. However, the dominant MRSA has gradually changed from ST239 to ST59 [29,38]. MRSA-ST59 is widely found in patient and food samples [3,18,20,21,37,38] and is becoming the predominant MRSA isolate of foodborne *S. aureus* and patient isolates in China.

## 5. Conclusions

This study has shown the resistance and prevalence characteristics of *S. aureus* isolated from patients and food in Shijiazhuang, China. Many MRSA isolates are distributed in patient isolates, while they are rarely detected in food samples. MRSA isolates had higher resistance to a variety of antibiotics in comparison to MSSA strains. MRSA-ST5, MRSA-ST59, and MRSA-ST239 were the predominant MRSA isolates in hospitals. The *S. aureus* isolates from patient and food samples have different characteristics. CC5, CC59, CC22, and CC398 were the most dominant CCs in patient specimens, while CC1, CC5, CC7 and CC59 were the epidemic CCs in food samples. The molecular characteristics can be used to describe *S. aureus* populations in infectious diseases and food contamination and provide evidence for the epidemic characteristics in transmissions. These data could help us understand the prevalence and resistance to antibiotics of this bacterium.

## Figures and Tables

**Figure 1 pathogens-11-01333-f001:**
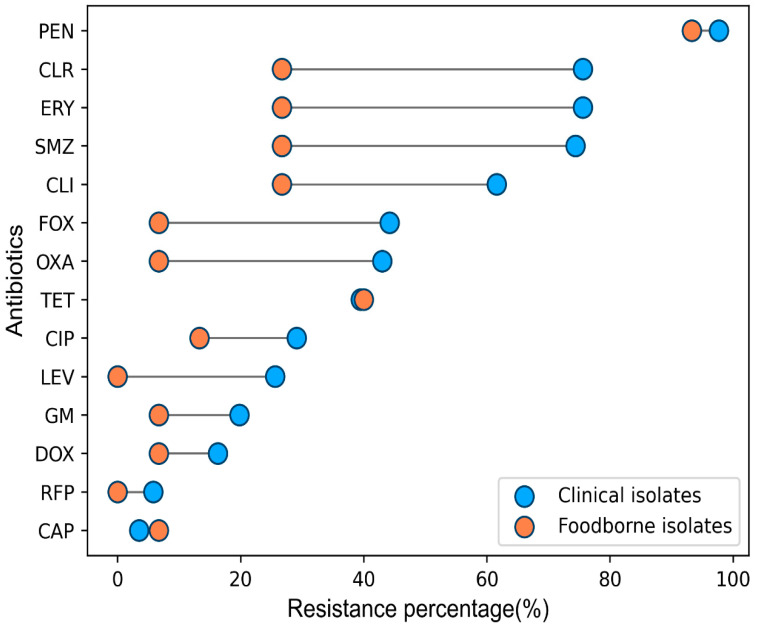
Antimicrobial resistance profiles of *S. aureus* isolated from patient and food samples to the same antibiotic. PEN penicillin, FOX cefoxitin, OXA oxacillin, TET tetracycline, DOX doxycycline, CAP chloramphenicol, RFP rifampicin, ERY erythromycin, CLR clarithromycin, SMZ sulfamethoxazole, GM gentamicin, CLI clindamycin, CIP ciprofloxacin, LEV levofloxacin.

**Figure 2 pathogens-11-01333-f002:**
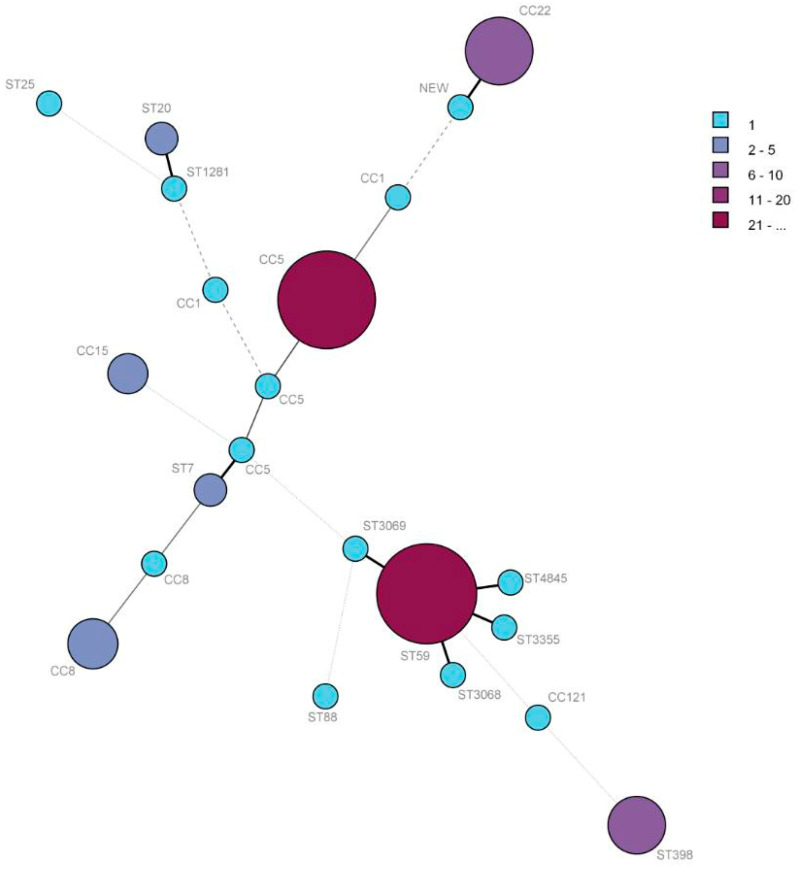
Phylogenetic tree of 86 patient *S. aureus* isolates based on MLST allelic profiles. Each circle represents CC type or singleton, and the size and color depth of each circle correspond to the number of samples (1, 2–5, 6–10, 11–20, and ≥21).

**Table 1 pathogens-11-01333-t001:** The antibiotics resistance ratio (%) of MRSA and MSSA.

Antimicrobial Agents	Patient Samples (*n* = 86) Resistance Rate (%)		
	Overall (*n* = 86)	MRSA (*n* = 38)	MSSA (*n* = 48)
PEN	84 (97.7%)	38 (100)	46 (95.8)
FOX	38 (44.2%)	38 (100) **	0 (0)
OXA	37 (43.0%)	37 (97.4) **	0 (0)
TET	34 (39.5%)	26 (68.4) **	8 (16.7)
DOX	14 (16.3%)	14 (36.8) **	0 (0)
CAP	3 (3.5%)	1 (2.6)	2 (4.2)
RFP	5 (5.8%)	4 (10.5)	1 (2.1)
ERY	65 (75.6%)	28 (73.7)	37 (77.1)
CLR	65 (75.6%)	28 (73.7)	37 (77.1)
SMZ	64 (74.4%)	26 (68.4)	38 (79.2)
GM	17 (19.8%)	9 (23.7)	8 (16.7)
CLI	53 (61.6%)	28 (73.7) *	25 (52.1)
CIP	25 (29.1%)	21 (55.3) **	4 (8.3)
LEV	22 (25.6%)	20 (52.6) **	2 (4.2)

PEN penicillin, FOX cefoxitin, OXA oxacillin, TET tetracycline, DOX doxycycline, CAP chloramphenicol, RFP rifampicin, ERY erythromycin, CLR clarithromycin, SMZ sulfamethoxazole, GM gentamicin, CLI clindamycin, CIP ciprofloxacin, LEV levofloxacin. * *p* < 0.05, ** *p* < 0.01.

**Table 2 pathogens-11-01333-t002:** Molecular characteristics of STs in patient and food *S. aureus*.

CCs	STs Number	Patient Samples					Food Samples
		Sputum	Pus and Secretion	Blood	Wound and Swabs	Others	Meats
CC1	ST1	0	0	0	0	0	4
	ST9	0	0	0	0	0	2
	ST2315	0	0	0	1	0	0
	ST6769	1	0	0	0	0	0
CC5	ST5	14	0	2	3	2	0
	ST6	0	0	0	0	1	2
	ST306	0	0	0	0	1	0
CC8	ST72	1	0	0	0	0	0
	ST239	2	1	0	0	2	0
CC15	ST15	1	0	0	0	2	1
CC22	ST22	2	0	0	3	5	0
CC121	ST121	0	0	0	0	1	0
ST7	ST7	2	0	0	0	0	2
ST20	ST20	1	1	0	0	0	0
ST25	ST25	0	1	0	0	0	0
ST59	ST59	6	8	1	2	5	2
ST88	ST88	0	0	0	0	1	0
ST398	ST398	1	2	1	0	3	0
ST522	ST522	0	0	0	0	0	1
ST1281	ST1281	0	0	0	0	1	0
ST2472	ST2472	0	0	0	0	0	1
ST3068	ST3068	1	0	0	0	0	0
ST3355	ST3355	1	0	0	0	0	0
NEW	NEW	1	0	0	0	0	0
ST3069	ST3069	0	0	0	1	0	0
ST4845	ST4845	0	0	0	1	0	0
Total		34	13	4	11	24	15

## Data Availability

Not applicable.

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
