# Peer review of "Molecular Characteristics and Antibiotic Resistance of Staphylococcus aureus Isolated from Patient and Food Samples in Shijiazhuang, China"

_pathogens, 2022, doi:10.3390/pathogens11111333_

Round 1

Reviewer 1 Report

The manuscript “Molecular Characteristics and Antibiotic Resistance of Staphylococcus aureus Isolated from Patient and Food Samples in Shijiazhuang, China” presents novel and very interesting results that contribute to the study of a bacterial species of great clinical importance. Therefore, this work deserves to be published in the journal, after the following minor corrections:

-        In lines 55-56 the MLST tool is mentioned, for what purpose is the MLST analysis performed? What information does it provide from the epidemiological point of view? What could this information be used for? Please, add this information.

-        In lines 114-115 the CLSI document (M100-S30) is mentioned. Please add its reference.

-        In lines 140-141 get the sentence “The antimicrobial resistance rates of the patients and food isolates were similar to the overall results (Figure.1)” out and replace it with the sentence of the lines 144-145 The difference in resistance rates between patients and food isolates to the same antibiotic was shown in Figure1.”

-        In lines 156-157, the sentence “However, there was no significant difference in resistance rates between patient and food isolates compared to the rest of the antibiotics.” is not understood. In this part of the results, I would talk about which antibiotics presented the same percentage of resistance in both foodborne and clinical strains.

-        In Figure 1, change “clinial” to “clinical”.

-        In line 195 instead of CC59 it would not be correct ST59?

-        In line 196 instead of CC398 it would not be correct ST398?

-        In figure 2, improve image resolution.

-        In lines 248-249 I don't understand the sentence. The correct CC would not be CC1, CC5 and CC22?

The results that are discussed in the lines 279-283 are not found in the results part. They should appear first in the results part to be discussed later.

Author Response

Thank you for your suggestions and comments! We have replied to your comments point by point and compiled them into a document for uploading. Please check it.

Reviewer 2 Report

The authors of this article present a study investigating the distribution of antibiotic resistance, MLST and CCs amongst S. aureus isolates from a region in China. While the experimental work is scientifically sound, I do have some concerns that are outlined below:

1. The biggest limitation of this study is that there were only 15 food isolates. You cannot make assertions about the results generated when a group size is smaller than 30. The larger the group size, the more likely the results are to be representative of the distribution within a population. 

2. The introduction is quite disjointed and needs to be improved. The sentences within each paragraph don't flow very well.

3. Methods section 2.1 - the % of each food type screened doesn't seem necessary. Also lines 83-84 sound more like a result than a method.

4. On line 100 there is an error - 0um instead of 10um in the brackets after blaZ-F.

5. The outcome of the blaZ and mec PCRs is not accurately represented in the results. Perhaps more detail in the text as well as a supplementary table would be useful.

6. The discussion is very confusing. There is a lot of repetition of the results followed by findings from other papers. Rather than repeating their findings and listing the findings of others there needs to be some comparison or discussion of the relevance of this to their own data.

7. The discussion has too many lists of CCs and STs. It's very difficult to follow. Rather than listing information, the authors need to make the relevant points about their data and how it relates to the findings of others. The discussion is the place where the results need to be interpreted.

8. Lines 247-260 - once again, due to the small group size, I think it's a little difficult to make assertions about the distribution of CCs and STs.

9. In the conclusion the authors state that their data "....explains the prevalence, transmission and resistance....". They have not supplied any evidence or discussion of transmission.

10. Lastly, the authors haven't made it very clear what the importance of this study was or what the significance of the findings were.

Author Response

(The authors gave the same response as above.)

Round 2

Reviewer 2 Report

Thank you for sending through a revised version of the manuscript. I understand that food borne isolates are quite uncommon and you are limited in the number available to you. However, having only 15 isolates is a limitation of this study. A sample size of at least 30 is required to make assertions about the distribution of your data. This doesn't mean that you can't publish this work, however this will need to be made clear at some point in the discussion. You can say that your data is consistent with that reported previously however due to small sample size, this may not be representative of true CC and ST distributions among food isolates.

Author Response

Dear reviewer:

Thank you very much for your comments on our revised manuscript. They are all valuable and very helpful for improving our discussion section and conclusion section in our paper. Revised portion are marked in the paper.

Due to the small foodborne sample size, we can’t make assertions. We removed the assertions statement in the discussion section. Instead, we make the relevant points about our data and how it relates to the findings of others. Accordingly, we also improved the conclusion section.

We appreciate for your warm work earnestly. Once again, thank you very much for your valuable advice.

Sincerely

Han Zhang